# Antioxidant and Anti-Inflammatory Functions of High-Density Lipoprotein in Type 1 and Type 2 Diabetes

**DOI:** 10.3390/antiox13010057

**Published:** 2023-12-28

**Authors:** Damien Denimal

**Affiliations:** 1Unit 1231, Center for Translational and Molecular Medicine, University of Burgundy, 21000 Dijon, France; damien.denimal@u-bourgogne.fr; 2Department of Clinical Biochemistry, Dijon Bourgogne University Hospital, 21079 Dijon, France

**Keywords:** HDL, oxidative stress, inflammation, diabetes, cardiovascular disease, paraoxonase

## Abstract

(1) Background: high-density lipoproteins (HDLs) exhibit antioxidant and anti-inflammatory properties that play an important role in preventing the development of atherosclerotic lesions and possibly also diabetes. In turn, both type 1 diabetes (T1D) and type 2 diabetes (T2D) are susceptible to having deleterious effects on these HDL functions. The objectives of the present review are to expound upon the antioxidant and anti-inflammatory functions of HDLs in both diabetes in the setting of atherosclerotic cardiovascular diseases and discuss the contributions of these HDL functions to the onset of diabetes. (2) Methods: this narrative review is based on the literature available from the PubMed database. (3) Results: several antioxidant functions of HDLs, such as paraoxonase-1 activity, are compromised in T2D, thereby facilitating the pro-atherogenic effects of oxidized low-density lipoproteins. In addition, HDLs exhibit diminished ability to inhibit pro-inflammatory pathways in the vessels of individuals with T2D. Although the literature is less extensive, recent evidence suggests defective antiatherogenic properties of HDL particles in T1D. Lastly, substantial evidence indicates that HDLs play a role in the onset of diabetes by modulating glucose metabolism. (4) Conclusions and perspectives: impaired HDL antioxidant and anti-inflammatory functions present intriguing targets for mitigating cardiovascular risk in individuals with diabetes. Further investigations are needed to clarify the influence of glycaemic control and nephropathy on HDL functionality in patients with T1D. Furthermore, exploring the effects on HDL functionality of novel antidiabetic drugs used in the management of T2D may provide intriguing insights for future research.

## 1. Introduction

According to the World Health Organization, approximately 422 million people worldwide are living with diabetes, of whom approximately 90% have type 2 diabetes (T2D) [1]. People with type 1 diabetes (T1D) or T2D have an excess risk of both fatal and non-fatal atherosclerotic cardiovascular events [2,3]. Dyslipidemia is a major contributor to the increased cardiovascular risk in these populations. Abnormalities in lipid metabolism in patients with T2D are mainly characterized by high serum triglycerides (≥150 mg/dL) and low levels of high-density lipoprotein cholesterol (HDL-C) (<40 mg/dL in men and <50 mg/dL in women) [4]. In T1D, only people with poor glycaemic control are most likely to have quantitative lipid abnormalities [5]. Beyond the quantitative aspect, lipoproteins also undergo qualitative abnormalities in both T1D and T2D that are relative to lipid and protein changes and chemical modifications.

In essence, HDLs, like all lipoproteins, are composed of apolipoproteins (apo), non-structural proteins, and lipids. The surface of HDLs is characterized by a monolayer composed of amphipathic phospholipids, sphingolipids, and unesterified cholesterol. This outer layer surrounds the particle core, containing a mixture of triglycerides and cholesteryl esters. HDLs exhibit heterogeneity regarding their size, density, and lipid and protein composition. Classically, these particles are referred to as HDL2a and 2b (larger), HDL3a, 3b, and 3c (medium or smaller), and lipid-free apoA-I or pre-β (discoidal). HDLs have multiple antiatherogenic properties that prevent the development of atherosclerotic lesions. The best-known effect of HDLs is their ability to promote cholesterol efflux from vascular cells and transport it for its elimination from the body. But HDLs exhibit many important antiatherogenic functions other than cholesterol efflux, such as anti-inflammatory and antioxidant effects or the ability to induce nitric oxide (NO) production by the endothelium. The interest in HDL functionality has increased over the past decade as several studies have demonstrated that measuring HDL functions outperforms serum HDL-C in cardiovascular risk stratification [6].

The interplay between HDLs and diabetes can be conceptualized as a bidirectional relationship [7]:On one hand, diabetes has detrimental effects on the composition and functions of HDLs, impairing, at least in part, their antiatherogenic properties [8,9,10]. Chronic hyperglycaemia, oxidative stress and inflammation are hallmarks of diabetes, fostering glycoxidative damage notably in lipoproteins, thereby detrimentally affecting the cardiovascular system [11,12]. Beyond the well-known glycoxidized low-density lipoproteins (LDLs), HDLs are also particles prone to glycoxidation (Figure 1).On the other hand, HDLs seem to fulfil functions on glucose metabolism that mitigate the development and progression of diabetes [13,14].

Therefore, within the context of the Special Issue of *Antioxidants* titled “Impact of antioxidant and inflammatory functions of HDL in diseases”, the primary objective of this narrative review is to report on the antioxidant and anti-inflammatory activities of HDLs in both T1D and T2D, contextualized within the framework of atherosclerotic cardiovascular disease. The secondary aim is to discuss the contribution of the antioxidant and anti-inflammatory functions of HDLs to the onset of diabetes.

## 2. Methods

This narrative review is based on a selection of research studies exploring the antioxidant and anti-inflammatory functions of HDLs in T1D and T2D within the context of cardiovascular diseases and the development of diabetes. Relevant English-language original articles, editorials and reviews published up to November 2023, were selected from the PubMed database. The search employed terms such as “HDLs”, “high-density lipoprotein”, “apoA-I”, “diabetes”, “antioxidant”, “antioxidative”, “anti-inflammatory”, and “antidiabetic” to identify articles deemed relevant for this review.

## 3. Antioxidant Functions of HDLs in Diabetes

Oxidative stress plays a pivotal role in the pathophysiology of both diabetes and atherosclerosis [11,15,16], stemming from an imbalance between the production of ROS and reactive nitrogen species and the antioxidant mechanisms [16]. ROS are produced as by-products of mitochondrial respiration or cellular metabolism, as well as through specific enzymes, including nicotinamide adenine dinucleotide phosphate (NADPH) oxidases. Ultimately, oxidative stress promotes endothelial dysfunction, pro-inflammatory pathways in the vascular wall and alterations in lipoproteins on both lipids and proteins. For example, it leads to the formation of oxidized LDLs (oxLDLs), which are widely acknowledged to be atherogenic [17]. HDLs, possessing antioxidant properties, serve as protective agents against damages induced by oxidative stress. The antioxidant functions of HDLs hold considerable relevance in the context of cardiovascular health, as it has been shown to be predictive of future cardiovascular events [18,19].

### 3.1. Overview of HDL Antioxidant Functions

Figure 2 summarizes the main antioxidant properties of HDLs. In brief, HDL particles protect against oxidative damage through a variety of mechanisms and several components [20]:HDLs protect LDL particles from oxidation caused by free radicals or copper ions. They remove and inactivate oxidized lipids from LDLs [21]. OxLDLs promote inflammation and oxidative stress in vessels through their binding to scavenger receptors. They facilitate the recruitment of monocytes, the release of pro-inflammatory cytokines from macrophages, the disruption of endothelial cell homeostasis and NO production, and ROS production [17]. Therefore, the HDL-mediated protection of LDLs from oxidation contributes to the anti-inflammatory and antioxidants effects of HDLs.HDLs stimulate the production of NO [22], which functions as an antioxidant by decreasing Fenton-type reactions, terminating radical chain reactions, and inhibiting oxidases. NO protects against NADPH oxidase-derived ROS production in the vascular endothelium by S-nitrosylating NADPH subunits [23]. As developed below, the anti-inflammatory effect of NO also contributes to the HDL-mediated decrease in ROS production.

The antioxidative potential of HDL particles derives from their protein and lipid components. HDLs transport multiple proteins that possess antioxidative properties. Thus, numerous HDL-associated apolipoproteins, lipid transfer proteins and enzymes have been demonstrated to play a role in the antioxidative capacity of HDLs. Firstly, apoA-I, the main protein in HDLs, neutralizes oxidized lipids [24]. More specifically, the methionine residues of apoA-I located at Positions 112 and 148 have the ability to convert lipid hydroperoxides into redox-inactive lipid hydroxides [24]. Although one study showed that HDLs from transgenic mice overexpressing apoA-II exhibited increased antioxidative properties [25], another study reported contradictory findings, potentially attributable to apoA-I displacement [26]. In addition, apoE may have a role in HDL antioxidant functions. HDLs isolated from apoA-I^−/−^ mice treated with an apoE3-expressing adenovirus exhibited a higher antioxidant capacity than those isolated from untreated apoA-I^−/−^ mice, although at a lesser extent than after treatment with apoA-I-expressing adenovirus [27]. Furthermore, apoM has the ability to bind oxidized phospholipids and enhance antioxidative activity of HDLs [28]. Secondly, cholesterol ester transfer protein (CETP), which serves to exchange neutral lipids between apoB-containing lipoproteins and HDLs, may also contribute to the antioxidant properties of HDLs since CETP addition to HDLs enhances their ability to inhibit LDL oxidation [29]. It is hypothesized that CETP aids the transfer of lipid peroxides towards HDLs. Thirdly, paraoxonase-1 (PON1), a protein exclusively associated with HDLs in the circulation, plays a crucial role in the antioxidant functions of HDLs [30]. It is an enzymatic protein, which hydrolyses a wide range of substrates, including lipid hydroperoxides that result from the conjugated diene free radicals on the later phase of the oxidation of fatty acids [30]. PON1 facilitates the breakdown of lipid peroxides into carboxylic acids rather than aldehydes or ketones, thereby preventing the uptake of oxLDLs by macrophages and smooth muscle cells in the arterial wall [30]. ApoE may play an indirect role in HDL-PON1 activity, as the binding of recombinant PON1 to HDL particles containing apoE increases its activity, albeit to a lesser extent than when it binds to HDLs containing apoA-I [31]. Overall, PON1 prevents oxidative damage to LDLs, endothelial cells, macrophages, and HDLs themselves [32]. PON1 holds particular clinical relevance, as low PON1 activity has been identified as a predictor of incident cardiovascular events [18,33]. Lastly, glutathione peroxidase, another antioxidant enzyme associated with HDLs, catalyzes the reduction in hydrogen peroxide and lipid peroxides.

The antioxidative activity of HDLs is also influenced by their lipid content. It exhibits a negative association with the HDL content in sphingomyelins, ceramides and triglycerides [34,35], and a positive correlation with the HDL content in sphingosine-1-phosphate (S1P), phosphatidic acid and phosphatidylserines [34,35]. Such influences may be due to changes in physical properties, such as the fluidity and net charges in the HDL surface monolayer [36]. In addition, S1P, which is a bioactive lipid mainly carried by HDLs, stimulates endothelial nitric oxide synthase (eNOS)-dependent NO production by binding to S1P receptors on vascular cells [34]. Lastly, plasmalogens, a class of phospholipids containing a fatty alcohol with a vinyl-ether bond at the sn-1 position, may play a role in the HDL-mediated eNOS activation and antioxidant effects [37,38].

The effectiveness of HDL subpopulations in preventing oxidative damage to LDLs increases when their size decreases [39], likely due to distinct proteomic and lipidomic profiles. For instance, the small dense HDL3 subpopulation is characterized by a higher total protein, PON1, apoM and S1P content, as well as a lower phospholipid content [40]. ApoE is predominantly found in HDL2 [40], suggesting a marginal contribution to the overall antioxidant function of HDLs. ApoA-I and apoA-II are evenly distributed among HDL subpopulations [40], thereby excluding it as a potential explanation for variations in the antioxidative properties across HDL subpopulations.

### 3.2. Effects of In Vitro Glycoxidation on HDL Antioxidant Functions

Glycoxidative damages in HDLs occurring in diabetes are likely to alter their antioxidant effects. In vitro oxidation of methionine residues of apoA-I attenuates the inactivation of lipid hydroperoxides [36]. Both glycation and glycoxidation of HDLs reduce PON1 activity [41,42,43], as well as their ability to inactivate lipid hydroperoxides in erythrocyte membranes [42]. Furthermore, glycated HDLs are less able to counteract the inhibitory effect of oxLDLs on vasorelaxation, potentially resulting from decreased antioxidant capacity of glycated HDLs [44]. Lastly, glycation of HDLs lowers their content in S1P through apoM dimerization [45,46], thereby impacting the apoM/S1P/eNOS pathway.

### 3.3. Antioxidant Functions of HDLs in Type 1 Diabetes

Few studies have explored the antioxidant activity of HDLs in individuals with T1D. The capacity of apoB-depleted serum to prevent LDLs from oxidation was reduced in 30 patients with T1D in comparison to 30 non-diabetic individuals, without any effect related to glycaemic control in this study [47]. In addition, our group showed that HDL particles from patients with T1D failed to reverse the inhibitory effect of oxLDLs on the relaxation of the aortic blood vessels, that may be attributed, at least partially, to a reduced capability of HDLs to counteract LDL oxidation [48]. PON1 activity was lower in 70 participants with T1D in comparison to 30 individuals without diabetes [49], while it was alike in another study that included smaller groups [47].

HDL-mediated NO production was reduced by 22% in 70 adolescents with T1D when compared to healthy controls [49]. This could potentially reduce the antioxidant capacity of HDLs given the crucial role played by NO in this process. The decrease in NO production induced by HDLs may be attributed to a lesser amount of the HDL3 fraction in S1P in this population [50].

Lastly, our group demonstrated a depletion of HDL2 from patients with T1D in plasmalogens [50]. This depletion could contribute to the reduction in the HDL antioxidant capacity in T1D, as plasmalogens are able to scavenge oxygen radicals [51] and play a role in the HDL-mediated stimulation of eNOS [37].

### 3.4. Antioxidant Functions of HDLs in Type 2 Diabetes

More information on the antioxidant functions of HDLs is available for T2D compared with T1D [8]. Morgantini et al. found that HDLs from 73 patients with T2D had altered capacity to protect LDLs from oxidation compared with 31 control subjects [52], that could be due to a lesser ability to remove lipid hydroperoxides [42]. In contrast, there was no decrease in the antioxidative activity of HDLs in a small cohort of 17 individuals with T2D compared to 17 controls [53]. Takata et al. recently demonstrated that the ability of HDLs to inactivate lipid hydroperoxides was associated with coronary plaque characteristics in 38 patients with T2D, in particular the lipid content [54].

The antioxidant capacity of HDLs exhibits variation among HDL subfractions, with smaller HDL subfractions demonstrating heightened antioxidant efficacy [39]. Small dense HDL3b and HDL3c showed a reduced ability to prevent LDLs from chemically induced oxidation in individuals with T2D, whereas HDL2 does not exhibit this effect [55]. In contrast, Gowri et al. found that HDL2, but not HDL3, were less potent to protect LDLs from macrophage-induced oxidation [56].

The inhibitory effect of HDLs on ROS production appears to be reduced in T2D. HDLs from patients with T2D are unable to reduce endothelial ROS production and NADPH oxidase activity in human aortic endothelial cells [57,58]. The impairment of HDL-induced NO production in T2D could be involved since NO reduces NADPH oxidase-derived ROS production in vascular endothelium [23].

The impairment of HDL antioxidant function in T2D may result from various alterations in HDL proteome, primarily related to PON1. PON1 activity is reduced in patients with T2D [42,59,60,61,62]. ApoA-I might be exchanged in HDLs for serum amyloid A (SAA), an acute-phase protein whose circulating levels increased during diabetes-related inflammation and oxidative stress [63]. As apoA-I is required for PON1 stability in HDL particles, this could lead to a reduction in PON1 activity [64]. In addition, PON1 shows higher levels of glycation in patients with T2D compared to non-diabetic individuals, and PON1 glycation decreases its activity [42,43]. Interestingly, there is a greater reduction in PON1 activity in women with T2D than in men, suggesting that this mechanism contributes, at least partially, to the lack of the protective effect of female gender against cardiovascular diseases in this population [65]. Overall, the reduction in PON1 activity in T2D appears to be of great importance, as it has recently been reported that PON1 activity improves the prediction of severe coronary artery disease in this population [66]. An elevated presence of oxidized apoA-II has been noted in individuals with T2D [67], yet its specific impact on the compromise of HDL functionality remains unclear.

Alterations in the antioxidant functions of HDLs in T2D may be attributable in part to alterations in their lipid composition. Several studies have found that triglyceride enrichment in HDL3 subfractions, as well as cholesteryl ester depletion, are associated with impaired antioxidative activity [55,68,69]. The substitution of cholesteryl esters with triglycerides in the HDL lipid core affects the conformation of apoA-I, potentially altering the accessibility of methionine residues to lipid hydroperoxides [70,71]. Otherwise, the depletion of plasmalogens in HDLs probably reduces their antioxidant capacity in T2D [72]. In addition, a recent study revealed that elevated levels of oxidized fatty acids in HDLs from individuals with T2D are associated with a reduced ability to inhibit ROS production [58]. Interestingly, the ex vivo treatment of HDLs from patients with T2D using an apoA-I mimetic peptide with the capability to bind oxidized fatty acids (D-4F) restored the antioxidant capacity of HDLs in individuals with T2D [58]. This suggests that oxidized fatty acids have a significant impact on the impairment of HDL antioxidative function in T2D. Lastly, the role of HDL-S1P in the alterations of HDL functions in T2D remains unclear. In one hand, the circulating concentrations of S1P and its carrier protein in HDLs (apoM) were often found to be reduced in patients with T2D [73,74,75,76,77,78]. On the other hand, the S1P content in HDLs was often found to be normal in this population [72,79,80].

## 4. Anti-Inflammatory Functions of HDLs in Diabetes

The anti-inflammatory function of HDLs holds significant importance for cardiovascular outcomes due to its predictive ability for future major adverse cardiovascular events. Thus, it predicts incident cardiac events in patients with myocardial infarction, regardless of serum HDL-C [81]. Low anti-inflammatory capacity of HDLs was also predictive of incident cardiovascular events in a study conducted in the general population [82]. A one-standard deviation increase in the anti-inflammatory capacity of HDLs resulted in a 26% reduction in the risk of incident atherosclerotic cardiovascular events after adjustment for serum HDL-C and other cardiovascular risk factors [82]. Integrating the anti-inflammatory capacity of HDLs into the Framingham risk score enhanced the predictive efficacy of the model, outperforming that of serum HDL-C [82].

### 4.1. Overview of HDL Anti-Inflammatory Functions

The classical IκB kinase (IKK)/IκB-α/NF-κB pathway plays a pivotal role in vascular inflammation, leading to heightened expression of chemokines and adhesion molecules, and release of pro-inflammatory cytokines. These mechanisms are involved in the recruitment of inflammatory cells into the subendothelial space, which is an early mechanism in development of atherosclerosis.

Figure 3 summarizes the main anti-inflammatory activities of HDLs. Firstly, HDLs downregulate the expression of chemokines and adhesion molecules. HDLs inhibit the production of the C-C motif chemokine ligand 2 (CCL2), also known as monocyte chemoattractant protein 1 (MCP-1), by vascular cells, thereby reducing the migration of a variety of immune cells, including monocytes, memory T lymphocytes, and natural killer cells [83]. In addition, HDLs reduce the expression of vascular cell adhesion molecule (VCAM)-1, intracellular adhesion molecule (ICAM)-1 and selectin-E on the surface of endothelial cells [72]. Furthermore, HDLs inhibit the release of pro-inflammatory cytokines, such as TNF-α and IL-6. Lastly, they reduce the inflammatory M1 macrophage phenotype [84], and the transcriptional repressor activating transcription factor 3 (ATF3) in macrophages may play a role in this phenomenon [85].

Some HDL components are particularly important for the anti-inflammatory effects of HDLs. HDLs inhibit the NF-κB pathway through several overlapping mechanisms following their interaction with the receptors SR-B1 and S1PR via apoA-I and S1P, respectively, and the transporters ABCA1 and ABCG1: cholesterol efflux, inhibition of toll-like receptors (TLR)4 [86] and TLR2 [87], upregulation of 3β-hydroxysteroid-δ24 reductase [88,89] and heme oxygenase-1 [89], eNOS activation [90], and subsequent S-nitrosylation [91]. Ultimately, HDLs prevent the translocation of NF-κB into the nucleus and its binding to DNA [86], leading to the inhibition of the gene expression of CCL2, pro-inflammatory cytokines, adhesion molecules, and also NADPH-oxidase (NOX2). Lastly, HDL-apoM binds to and neutralises lipopolysaccharides, leading to a reduction in the release of TNF-α from macrophages [92]. The larger HDL subpopulations are less potent at inhibiting the expression of adhesion molecules compared with smaller HDLs [93,94], likely due, at least in part, to a lower content of apoM and S1P. Lastly, apoE may play a role in the anti-inflammatory properties of HDLs. It has been recently found that apoE-containing HDLs promote the survival of the anti-inflammatory regulatory T lymphocytes [95]. Interestingly, apoE-containing HDL-C is inversely associated with coronary artery calcium score and stenosis [96].

### 4.2. Effects of In Vitro Glycoxidation on HDL Anti-Inflammatory Functions

Glycoxidation of HDLs appears to modulate HDL anti-inflammatory functions. Firstly, the ex vivo treatment of plasma with L-4F, an apoA-I mimetic peptide capable of binding oxidized lipids with higher affinity than apoA-I itself, restores the anti-inflammatory function of HDLs [52,97]. Secondly, apoA-I modified by AGEs exhibits a reduced capacity, in comparison to native apoA-I, to inhibit the expression of the integrin CD11b, which is located on monocytes and interacts with adhesion molecule ICAM-1 [98]. Lastly, the in vitro glycation of HDLs using glycolaldehyde and malondialdehyde attenuated their ability to suppress the gene expression of SAA and CCL2 [99]. The in vitro glycation of apoA-I or HDLs reduces their ability to inhibit the release of pro-inflammatory cytokines, such as TNF-α and IL-1β, by macrophages following lipopolysaccharide stimulation [100,101].

In addition, glycated apoA-I exhibits a partial reduction in its capacity to inhibit cytosolic IκB-α phosphorylation and nuclear translocation of the NF-κB p65 subunit [100]. It may be attributable to a lower affinity of glycated apoA-I to macrophages compared to native apoA-I [100]. Beyond the effects of glycoxidation of apoA-I or HDLs on the NF-κB pathway, the carbamylation of HDLs also increases the phosphorylation of NF-κB in endothelial cells, thus facilitating its translocation to nuclei [102].

### 4.3. Anti-Inflammatory Functions of HDLs in Type 1 Diabetes

Very limited data on HDL anti-inflammatory functions are available in T1D. HDLs from patients with poorly controlled T1D kept their ability to suppress the gene expression of SAA and CCL2 in palmitate-stimulated 3T3-L1 adipocytes [99]. In addition, it has been shown that total HDLs from patients with T1D had the inhibitory effect on the endothelial expression of adhesion molecules that is similar to that of healthy controls [93]. In contrast, medium dense HDL fraction and, at a lesser extent, dense HDL fraction were less efficient to reduce VCAM-1 expression on endothelial cells in female T1D patients than in female controls, without any difference regarding E-selectin expression [93]. In this study, apoM and S1P were found to be partially moved from dense to light density HDL particles during T1D, but surprisingly without improving their anti-inflammatory effect [93].

The recent study by Chiesa et al. has provided evidence indicating that HDLs isolated from patients with T1D exhibit a reduced capacity to stimulate NO production by endothelial cells [49]. This mechanism likely contributes to the defect in HDL anti-inflammatory function regarding the inhibitory effect of NO on the inflammatory NF-κB pathway.

### 4.4. Anti-Inflammatory Functions of HDLs in Type 2 Diabetes

HDLs from patients with T2D have an impaired ability to impede the migration of monocytes towards endothelial cells [52,102]. Interestingly, this functional impairment was correlated with the plasma concentration of SAA and carbamylated HDLs [52,102]. In vitro carbamylation of HDLs replicates the observed decline in the ability to inhibit the migration of monocytes, thereby highlighting the potential involvement of carbamylated HDLs in this process [102]. The level of glycaemic control does not appear important in this loss of the HDL function [52].

As previously mentioned, HDL particles can reduce the expression of ICAM-1 and VCAM-1 on endothelial cells, thereby downregulating the recruitment of immune cells in the vascular wall [72]. HDLs from patients with T2D exhibit a diminished ability to inhibit the gene expression of VCAM-1 compared to those from healthy individuals [62,103]. The underlying mechanisms are not well understood, but glycoxidative modifications of HDL particles could be involved. This is suggested by the positive correlation between the reduction in HDL-induced VCAM expression, plasma fasting glucose, and the negative correlation with PON1 activity. In addition, the shift in HDL size towards the smaller HDL3 particles in T2D may also play a role, as HDL3 has demonstrated more potent anti-inflammatory activity than HDL2 [94]. The HDL/apoM/S1P pathway demonstrates notable efficacy in suppressing the expression of adhesion molecules [104]. Although apoM appears decreased in HDLs from patients with T2D [105], several groups have reported a conserved level of S1P in HDLs from patients with T2D [72,79,80]. Therefore, it is unlikely that the apoM/S1P axis plays a major role in this defect when it is observed. Interestingly, carbamylation of HDLs also results in the same functional defect [102], and carbamylated HDLs are predictive of all-cause mortality as well as cardiovascular-specific mortality in patients with T2D [106]. On the contrary, our group showed no impairment in the capability of HDLs to inhibit VCAM-1 and ICAM-1 expression in patients with T2D, despite classical alterations in HDL lipid composition such as triglyceride enrichment and esterified cholesterol depletion [72].

Otherwise, apoA-I or HDLs isolated from patients with T2D have a reduced capacity to inhibit the release of pro-inflammatory cytokines such as TNF-α and IL-1β by macrophages after lipopolysaccharide stimulation [100,101,107]. The effectiveness of HDL-induced inhibition of TNF-α release diminishes with the degree of impaired kidney function in patients with T2D [107]. In vitro enrichment of HDLs with SAA results in similar impairments [107]. The reduction in sphingomyelin content in HDLs from patients with T2D, observed by some research groups [105,108,109] (though not universally supported [72]), may also contribute to this deficiency in HDLs. In fact, reconstituted HDLs (rHDLs) containing sphingomyelins exhibited a greater capacity than those containing phosphatidylcholines to inhibit the production of pro-inflammatory cytokines [110].

From a mechanistic point of view, HDLs isolated from patients with T2D lose their capacity to inhibit the activating phosphorylation of NF-κB in endothelial cells [74]. It has been demonstrated that HDLs from patients with T2D are less able to stimulate the activating-phosphorylation of eNOS at serine 1177 [74], NO production [57], and vessel relaxation [57]. Such impairment in HDL-induced NO production may be involved in the reduction of anti-inflammatory properties of HDLs, as NO acts as an inhibitor of the NF-κB pathway in vascular cells [23]. Lastly, given that oxLDLs promote inflammatory pathways within the vasculature, the compromised ability of HDLs to protect LDLs from oxidation contribute to the attenuated anti-inflammatory functions of HDLs observed in patients with T2D [52].

## 5. Antioxidant and Anti-Inflammatory Functions of HDLs in the Development of Diabetes

Substantial evidence indicates that HDLs play a role in the development and progression of diabetes. Firstly, epidemiological studies demonstrated an inverse association between the development of T2D and HDL-C levels [111,112,113,114]. Elevated triglyceride to HDL-C and apoCIII to apoA-I ratios are indicative of a higher risk for the onset of T2D [115,116]. These observations have prompted the inclusion of HDL-C levels in risk assessment scores for T2D [117]. Nevertheless, the predictive value of apoA-I levels for the onset of T2D was not observed in individuals with prediabetes [118]. Interestingly, the variability in serum lipid levels, such as elevated triglycerides/HDL-C variability, has been linked to an increased risk of new-onset diabetes in Chinese adults [119]. In the context of T1D, the AMORIS study uncovered that apoA-I levels, as well as triglycerides and apoB/apoA-I ratio, serve as predictors for incident T1D in a cohort of 591,239 individuals in Sweden [120]. Secondly, Mendelian randomization studies have demonstrated that an elevation in serum HDL-C is associated with a reduced risk of developing T2D [121,122,123]. Thirdly, in large interventional studies, CETP inhibitors, which increased HDL-C concentrations by 30 to 130%, demonstrated an average 16% reduction in the risk of new-onset diabetes [124]. Finally, many antidiabetic mechanisms triggered by HDLs or apoA-I have been identified [13]. As summarized in Figure 4, HDLs exert their influence on glucose metabolism by affecting various key components, especially the pancreas, skeletal muscle and adipose tissue.

### 5.1. HDLs and Pancreatic β-Cells

ApoA-I stimulates the expression of key enzymes involved in insulin maturation within pancreatic β-cells [125]. It increases the release of insulin by Ins-1E cells through a forkhead box protein O1 (FoxO1)-dependent mechanism [126], and HDLs stimulate insulin production in Min6 cells in a calcium dependent manner [127]. Infusions of rHDLs or apoA-I stimulated insulin secretion, resulting in decreased plasma glucose concentrations in both individuals with T2D and obese mice [128,129]. HDL-associated S1P may play a crucial role, as evidenced by the correlation between the S1P content of HDLs from patients with T2D and the capacity of HDLs to enhance insulin secretion by β-cells [130]. Furthermore, HDLs protect β-cells from apoptosis triggered by endoplasmic reticulum stressors [131].

Excessive ROS production contributes to β cell dysfunction and apoptosis, ultimately impairing insulin production and secretion. HDLs demonstrate the capability to scavenge ROS, thereby mitigating oxidative damage to cellular components and safeguarding β-cell integrity. Given its protective role against oxidative stress, PON1 is believed to positively correlate with insulin release by β cells. Administration of recombinant PON1 to mice prior to streptozotocin-induced diabetes resulted in a decreased incidence of diabetes and higher circulating insulin levels [132]. In cultured β cells, the reduction in oxidative stress induced by PON1 was inversely associated with the ability of cells to release insulin [132].

### 5.2. HDLs and Muscle Cells

Both HDLs and apoA-I enhance glucose uptake in skeletal muscle cells, acting independently of insulin secretion stimulation. Administration of recombinant apoA-I in mice enhances the glucose uptake in skeletal muscle [133]. HDLs and apoA-I stimulate the insulin-independent AMP-activated protein kinase (AMPK) pathway, and apoA-I activates the insulin receptor pathway. Ultimately, the expression of glucose transporter type 4 (GLUT4) is enhanced at the cell surface. Mice deficient in PON1 exhibited insulin resistance [134]. PON1 upregulates GLUT4 expression in myotubes by enhancing the PI3K/Akt signalling pathway [134]. The antioxidative function of PON1 may also be accountable for the increased GLUT4 expression, as blocking the PON1 SH group at position Cys284 significantly diminishes its impact on GLUT4 expression compared to wild-type PON1 [134].

### 5.3. HDLs and Adipocytes

HDLs promote glucose uptake by 3T3-L1 adipocytes, stimulate the activation of Akt, and facilitate the translocation of GLUT4 to the cell membrane [135]. The contribution of the antioxidant and anti-inflammatory functions of HDLs to this process remains a subject of significant interest.

## 6. Effects of Diabetes Management on Antioxidant and Anti-Inflammatory Functions of HDLs

### 6.1. Lifestyle Interventions

Lifestyle interventions are effective strategies to prevent T2D and improve metabolic control in patients with T2D [136,137,138]. Surprisingly, the DPPOS study recently demonstrated that lifestyle intervention does not reduce the incidence of major adverse cardiovascular events in 3234 participants with impaired glucose tolerance followed for 21 years on average [138]. However, the intensive lifestyle intervention was implemented for a duration of only 3 years in this study [138]. On the contrary, the Da Qing Diabetes Prevention Outcomes Study showed that a 6-year lifestyle intervention resulted in a 26% reduction in the incidence rate of cardiovascular events during a 30-year follow-up in individuals with impaired glucose tolerance [139].

Physical activity is typically associated with an increase in HDL-C, as well as a decrease in serum LDL-C and triglycerides [140]. However, a Cochrane meta-analysis showed that exercise had no effect on serum HDL-C in patients with T2D [141]. Regarding HDL functionality, the capacity of the small HDL3c fraction to prevent LDL oxidation was improved in patients with metabolic syndrome following a 12-week educational program centered on a reduction in caloric intake and an elevation in physical activity [142]. HDL2a and HDL3b were more efficient in preventing LDLs from oxidation after 3 months of moderate intensity training in 30 subjects with metabolic syndrome [143]. In a small cohort of 11 patients with T2D, aerobic training improved by 15% the effect of HDL3 against LDL oxidation [144]. Lastly, PON1 activity in serum and HDL3 increased after a 10-week walk/run training program, in relation with a reduction in serum malondialdehyde [145].

Smoking cessation elicited a modest yet prompt elevation in both serum HDL-C and apoA-I within a cohort comprising 65 smokers [146]. In addition, the ability of HDLs to inactivate oxidized phospholipids improved after smoking cessation [147]. This improvement was associated with a reduction in the malondialdehyde content of HDLs, a by-product resulting from lipid peroxidation [147].

Regarding the effects of lifestyle interventions on HDL anti-inflammatory function, HDL3 were more efficient in downregulating the expression of VCAM-1 on endothelial cells and the release of CCL2 after a 10-week walk/run training program in a study population not strictly recruited based on the presence of diabetes but rather on the presence of metabolic syndrome [145]. A short-term diet and exercise intervention enhanced the capacity of HDLs to reduce monocyte chemotactic activity in a study that enrolled men based on the presence of overweight or obesity rather than strictly on the presence of T2D [148].

Diet interventions may improve antioxidant functions of HDLs. Supplementation with ginger or *Salvia miltiorrhiza* extract improved PON1 activity in patients with T2D [149,150]. The arylesterase activity of PON1 and LCAT activity in HDL3 fractions from individuals with T2D increased after a diet enriched with fruit and vegetable for 8 weeks [151].

### 6.2. Bariatric Surgery

Bariatric surgery increases serum HDL-C in T2D [152], and appears to be beneficial for HDL functions. Their ability to prevent LDL oxidation was improved at 6 months after bariatric surgery compared to before surgery in obese patients, while HDLs were less susceptible to oxidation [153]. Similar conclusions were yielded at one year after sleeve gastrectomy in adolescent males [154]. Primarily, Osto et al. demonstrated that Roux-en-Y gastric bypass improved the ability of HDLs to reduce endothelial NADPH oxidase activity at 12 weeks after surgery in 29 patients [155]. In this study, PON1 activity increased after surgery, as well as the capacity of HDLs to stimulate endothelial NO production and reduce VCAM-1 expression [155].

### 6.3. Glucose Control Agents

Longitudinal studies, such as the UK Prospective Diabetes Study (UKPDS) and the Diabetes Control and Complications Trial (DCCT), have provided evidence that medications such as insulin and metformin, which enhance glucose control, are linked to a reduction in chronic complications. Additionally, though not extensively reported, they may be associated with lower levels of glycated lipoproteins. Certain medications, like metformin, may also exhibit pleiotropic effects, such as antioxidant or anti-AGE effects. These effects are likely attributed to the impact of these medications on reducing glucose levels, as well as related improvements in the lipid profile and other pleiotropic factors. Our group recently demonstrated that the improvement of glycemic control through standard care in a cohort of 27 patients with T1D resulted in a reduction in the carbamylation of HDLs [156]. This is noteworthy as carbamylated HDLs exhibit diminished antioxidant and anti-inflammatory properties [157].

Limited data exist regarding the impact of antidiabetic agents on the antioxidant and anti-inflammatory functions of HDLs. In vitro incubation of metformin with glycated HDLs reduced the formation of AGEs in HDLs [158] that could potentially improve their antiatherogenic functions. The capacity of HDLs to decrease endothelial VCAM-1 expression exhibited greater efficacity in rats treated with liraglutide, a glucagon-like peptide 1 receptor agonist [155]. This effect could be attributed, at least partially, to the heightened ability of HDLs to induce NO production following liraglutide administration [155]. Liraglutide treatment did not enhance PON1 activity in rats, but it reduced NADPH oxidase activity in endothelial cells and decreased superoxide anion levels in rat aortas [155]. Concerning thiazolidinediones, our research team observed no enhancement in the vasorelaxant effect of HDLs in individuals with T2D treated with rosiglitazone or pioglitazone, despite a rise in HDL-C levels [159].

Sodium glucose co-transporter 2 (SGLT2)-inhibitors have shown great benefit for reducing major adverse cardiovascular events in people with diabetes [160]. A recent meta-analysis including 41,320 individuals found that SGLT2-inhibitor treatment slightly increased HDL-C by 0.06 mmol/L (2.23 mg/dL) [161]. However, there are currently no published studies examining their effects on HDL glycoxidation. Dapagliflozin downregulated NADPH oxidase-dependent ROS production and malondialdehyde levels in mouse cardiac tissues [162], while canagliflozin reduces NADPH oxidase expression and lipid peroxides in the kidneys of diabetic rats [163]. But dapagliflozin treatment did not improve PON1 activity after 12 weeks in 15 patients with T2D [164], as well as HDL-mediated endothelial NO production after 4 weeks in eight patients with both T2D and CAD [165]. Further investigations with larger study populations are required to draw conclusive findings.

### 6.4. Statins and Fibrates

Regarding the effects of lipid-lowering agents, rosuvastatin corrects HDL-apoA-I kinetics abnormalities in patients with T2D, with an effect on both catabolism and production rate [166]. It decreases the levels of oxidized HDLs and improves PON1 activity in men with T2D treated by 20 mg/day orally for a period of 12 weeks [167]. Administration of simvastatin (40 mg/day) over an 8-week period in 14 men with T2D did not result in an enhancement of the in vitro capability of HDLs to inhibit LDL oxidation and CCL2 expression by endothelial cells [168]. Similar results have been observed for bezafibrate [168].

### 6.5. Omega-3 Fatty Acids

The JELIS and REDUCE-IT trials demonstrated a significant reduction in cardiovascular events in patients treated with omega-3 fatty acids, while the STRENGTH trial did not [169,170,171]. In a subset of the JELIS trial, specifically conducted in hypercholesterolemic patients with impaired glucose metabolism, the arm receiving statins plus 1.8 g/day of eicosapentaenoic acid (EPA) exhibited a substantial 22% reduction in the incidence of coronary artery disease compared to the arm receiving statins alone [169]. It is well documented that omega-3 fatty acids reduce triglyceride levels in patients with T2D [172], but they also exert pleiotropic effects on multiple atherosclerotic processes including oxidative stress and inflammation [173]. The influence of EPA on HDL functions has been poorly studied. Treatment with 2 g/day EPA for 8 weeks increased serum levels and activity of PON1 in a small randomized clinical trial involving 36 patients with T2D compared to placebo [174]. In contrast, a 6-week intervention with 2 g EPA and docosahexanoic acid (DHA) supplement did not improve serum PON1 activity in patients with T2D [175]. A recent study further elucidated the antioxidant effects of EPA by comparing the effects of placebo, EPA, and DHA on the oxidation rates of small-dense LDL, very-low-density lipoproteins, and membranes [176]. EPA exhibited potent and sustained antioxidant effects over time, distinguishing itself from the effects observed with DHA and the placebo [176].

## 7. Effects of Treatment with HDL Mimetics on Oxidative Stress and Inflammation

The landscape of HDL-based therapeutics has shifted its focus from merely increasing circulating HDL-C to enhancing HDL functionality. HDL mimetics, consisting of nanoparticles engineered with apoA-I as the scaffold, present a versatile range of therapeutic properties by augmenting HDL quality. Current evidence from both pre-clinical and clinical studies supports the notion that augmenting cholesterol efflux through HDL mimetics is a viable strategy, contributing to the stabilization of atherosclerotic plaques via various immunoregulatory processes [177]. Table 1 provides key data outlining the anti-inflammatory and antioxidant effects of HDL mimetics that are pertinent to cardiovascular health.

Infusion of CSL-111 into mice fed a Western-type diet has anti-inflammatory effects in lesion macrophages [182]. In patients with T2D, CSL-111 increases the ex vivo capability of HDLs to inhibit the expression of adhesion molecules on endothelial cells and decrease the expression of CD11b on leukocytes [181]. CSL-112, composed of two molecules of apoA-I and 110 molecules of phosphatidylcholine, appears as one of the most promising HDL mimetics [191,192]. CSL-112 particles were rapidly remodelled after infusion in humans, resulting in particular in the formation of small highly functional HDL particles [187]. They were more efficient than native HDL3 to reduce the release of IL-6, IL-1β and TNF-α by peripheral blood mononuclear cells, and were as effective as HDL3 in inactivating lipid peroxides in LDLs [187]. The ongoing phase 3 AEGIS-II trial (NCT03473223) is currently investigating the efficacity of a 4-week treatment with CSL-112 on cardiovascular events in high-risk patients with acute myocardial infarction [193].

## 8. Discussion

Oxidative stress and inflammation are intricated mechanisms implicated in diabetes and the progression of atherosclerotic lesions [194]. The antioxidant and anti-inflammatory properties of HDLs play a crucial role in mitigating the adverse effects of oxidative stress and inflammation in diabetes. The present review adds to existing literature by highlighting new insights on the HDL functionality in both T1D and T2D [9,10], with particular emphasis on the intricate interplay between compositional alterations and antiatherogenic functions of HDLs. In addition, this review underscores the detrimental impact of glycoxidation on the composition and functions of HDLs.

Table 2 summarizes the findings from the current review concerning the antioxidant and anti-inflammatory functions of HDLs in both T1D and T2D. In patients with T2D, the antioxidant functions of HDLs on LDLs and ROS production are impaired [52,57,58], with a notable contribution of the alterations in PON1 activity in this population [42,59,60,61,62]. Limited studies in patients with T1D hinder definitive conclusions regarding the antioxidant functions of HDLs in this specific population.

The anti-inflammatory capacities of HDLs are compromised in T2D, characterized by a reduced capacity to impede the migration of monocytes into the subendothelial space and the release of pro-inflammatory cytokines [52,102]. Alterations in HDL-induced NO production in T2D may play a pivotal role [57]. Data regarding the ability of HDLs to downregulate the endothelial expression of adhesion molecules appear more nuanced [62,72]. Although limited results are available in T1D, recent findings revealed a reduction in HDL-induced NO production in patients with T1D [49], suggesting a potential impact on HDL anti-inflammatory functions.

The HDL efflux capacity may contribute to the impairment of the antioxidant and anti-inflammatory properties of HDLs in diabetes, notably through the effects of HDLs on NO production. The binding of HDLs to SR-BI via apoA-I causes cholesterol efflux that is sensed by SR-BI, leading to PDZK1-dependent activation of Src family kinases and Akt, which phosphorylates eNOS at Ser1177 and thereby increases NO production [196,197,198]. In individuals with T1D, apoB-depleted serum cholesterol efflux has been reported to be increased using J774 macrophages [195], although contradictory findings were observed in a smaller study cohort [47]. The plasma cholesterol efflux of 14 individuals with T1D was increased using Fu5AH cells and fibroblasts [199]. Our group recently conducted a comprehensive review of data on HDL cholesterol efflux in T2D. The results exhibit a high degree of heterogeneity, preventing the formulation of definitive conclusions [8].

HDLs exert multifaceted roles in diabetes development, with their antioxidant and anti-inflammatory functions acting as pivotal guardians against oxidative stress and chronic inflammation. HDL antioxidant functions seem to contribute to preventing β cell dysfunction related to oxidative stress. PON1 emerges as a crucial determinant of this beneficial effect, as well as in the capacity of HDLs to facilitate glucose uptake in muscle cells.

Some gaps persist in elucidating the pathophysiology of the antiatherogenic functions of HDLs in diabetes. For instance, the capacity of HDLs to inactivate lipid hydroperoxides and PON1 activity remains unclear in T1D. Additionally, considering the strong association of glycaemic control and nephropathy with cardiovascular risk in this population, it would be interesting to better understand the impact of these variables on HDL functionality. In the context of T2D, the findings regarding the ability of HDLs to reduce the expression of adhesion molecules are contrasted, warranting further investigation.

Cardiovascular risk stratification in patients with T2D remains a significant challenge. The assessment of HDL functions could a valuable approach to improve cardiovascular risk evaluation in this population. Several aspects related to HDLs, including HDL antioxidant and anti-inflammatory functions, as well as cholesterol efflux capacity and HDL size, have been shown to possess predictive value for incident cardiovascular events in the general population and in high-risk study populations [18,19,82,200,201,202,203]. However, the question remains unresolved specifically within the population of patients with T2D. In this regard, the innovative high-throughput cell-free assays designed for the evaluation of the antiatherogenic functions of HDLs represent promising tools for conducting such epidemiological studies [6].

Finally, from a therapeutic perspective, unravelling the impact of HDL mimetics on HDL functionality may provide novel insights into therapeutic strategies for diabetes and its associated complications. Interventions aimed at enhancing PON1 activity hold promise and are of particular clinical relevance in the diabetic population. Examining the impact of novel antidiabetic drugs, such as SGLT2-inhibitors, on HDL functionality may provide intriguing insights for future research.

## 9. Conclusions

Accumulating evidence has demonstrated an impairment of HDL antioxidant and anti-inflammatory functions in individuals with T2D. In contrast, data in T1D are less conclusive. A better understanding of these functions is crucial to identify intriguing targets to mitigate cardiovascular risk in these populations.

## Figures and Tables

**Figure 1 antioxidants-13-00057-f001:**
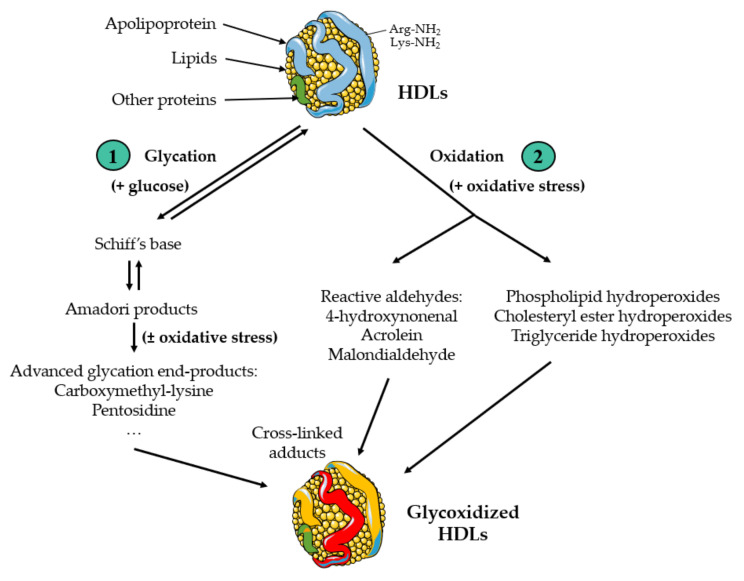
Glycation and oxidation of HDL particles. (1) Glycation of HDLs. Reducing sugars, like glucose, react with amino groups on HDL proteins, forming a reversible Schiff’s base and subsequently an Amadori product after rearrangement. These early glycation products can lead to the generation of advanced glycation end-products (AGEs), constituting a diverse family of derivatives. Oxidative stress further facilitates the formation of numerous AGEs, including carboxymethyl-lysine and pentosidine, which may be regarded as glycoxidation products. (2) Oxidation of HDLs. Reactive oxygen species (ROS) induce the oxidation of fatty acyl chains, encompassing those within phospholipids, esterified cholesterol, and triglycerides, leading to the generation of lipid hydroperoxides. ROS initiate the production of reactive aldehydes from polyunsaturated fatty acids. In particular, 4-hydroxynonenal, acrolein, and malondialdehyde are by-products of lipid peroxidation. These compounds have the capacity to modify proteins in HDLs through the formation of cross-linked adducts.

**Figure 2 antioxidants-13-00057-f002:**
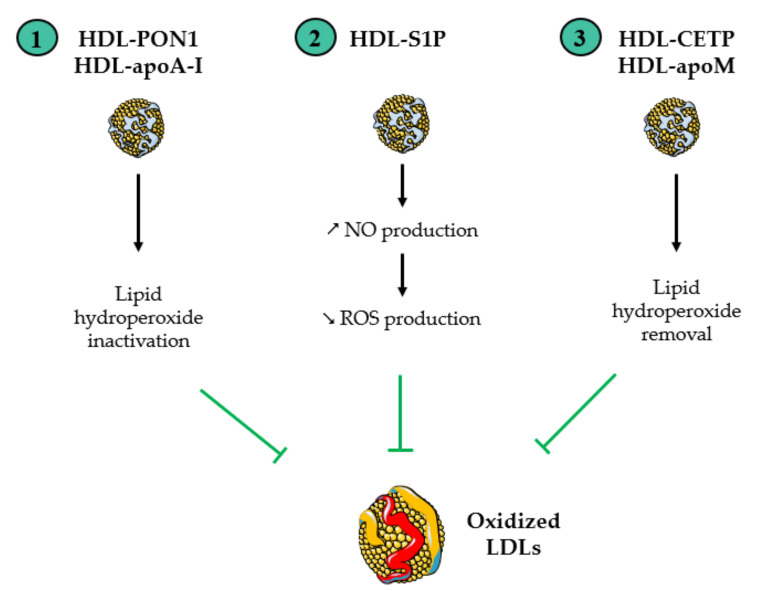
Antioxidant functions of HDL particles. “↗” and “↘” mean “increase” and “decrease”, respectively. HDLs protect LDLs from oxidation by several mechanisms. (1) HDL proteins paraoxonase-1 (PON1) and apoA-I inactivate lipid hydroperoxides in oxidized LDLs. (2) HDL-sphingosine-1-phosphate (S1P) stimulates the production of NO, which inhibits the formation of ROS by NADPH oxidase in immune cells. (3) It has also been suggested that the cholesterol ester transfer protein (CETP) and apoM may bind oxidized lipids and facilitate their removal from oxidized LDLs. The figure was partly created utilizing Servier Medical Art (Creative Commons Attribution 3.0 unported license).

**Figure 3 antioxidants-13-00057-f003:**
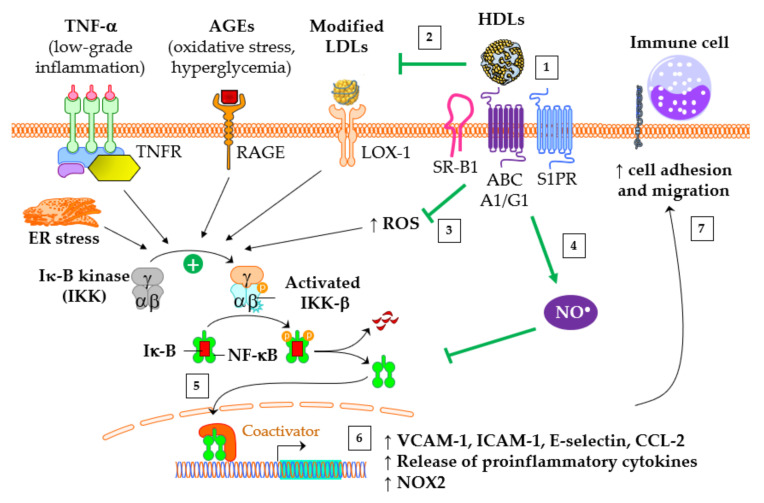
Main anti-inflammatory properties of HDLs. “↗” means “increase”. Several mediators promote the inflammatory NF-κB pathway in vascular cells, including endoplasmic reticulum (ER) stress, TNF-α, AGEs, ROS, and modified LDLs. This results in an elevated gene expression of adhesion molecules, pro-inflammatory cytokines, and NADPH oxidase (NOX2). (1) Healthy HDLs bind to S1P receptors (S1PR), SR-BI, ABCA1 and ABCG1. (2) HDL-mediated protection of LDLs against oxidation counteracts the activating effect of oxidized LDLs on NF-κB pathway through the binding to the LOX-1 receptor. (3) HDLs dampen ROS production. (4) The activation of endothelial NO synthase by HDLs leads to the inhibiting nitrosylation of NF-κB. (5) Ultimately, HDLs have the capacity to impede the translocation of NF-κB into the nucleus, (6) subsequently suppressing the gene expression of pro-inflammatory cytokines and adhesion molecules. (7) Such transcriptional effects inhibit the recruitment of immune cells into the subendothelial space. The figure was partly created utilizing Servier Medical Art (Creative Commons Attribution 3.0 unported license).

**Figure 4 antioxidants-13-00057-f004:**
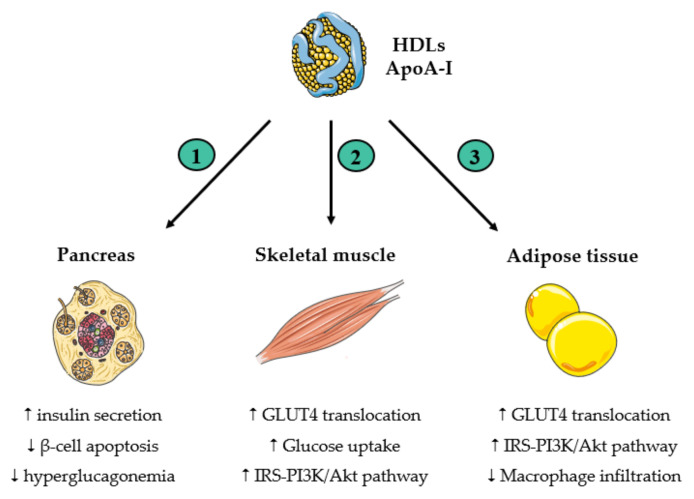
Antidiabetic properties of HDLs and apoA-I. “↑” and “↓” mean “increase” and “decrease”, respectively. (1) In pancreas, HDLs increase insulin secretion, reduce β-cell apoptosis, and counteract glucagon overproduction in α-cells. (2) In skeletal muscle cells, HDLs increase glucose uptake through the insulin receptor substrate (IRS)-1/phosphatidyl inositol 3-kinase (PI3K)/Akt signalling pathway. (3) In adipose tissue, HDLs activate IRS-1-PI3K/Akt pathway, facilitate the translocation of GLUT4 to the cell membrane, and reduce macrophage infiltration. The figure was partly created utilizing Servier Medical Art (Creative Commons Attribution 3.0 unported license).

**Table 1 antioxidants-13-00057-t001:** Effects of HDL mimetics on oxidative stress and inflammation.

Reference	HDL Mimetic	Model	Findings
[178]	Ac-hE18A-NH2	HUVECs THP-1	↘ VCAM-1 expression ↘ monocyte adhesion ↘ IL-6 release ↘ CCL2 release
[179]	CSL-111	HMDMs	↘ NF-κB activation ↘ TNF-α, IL-6, and IL-1β
[180]		Wild-type mice Patients with T2D	↘ CD11b on monocytes ↘ leukocytes
[181]		Patients with T2D	↗ HDL ability to inhibit VCAM-1 and ICAM-1 expression on HCAECs
[182]		Wild-type mice LDLR^−/−^ mice apoE^−/−^ mice	↘ IL-6, TNF-α, and CCL2 expression by macrophages No effect on IL-6, TNF-α, and CCL2 ↘ IL-6 and IL-1β expression on aortic macrophages and neutrophils
[183]	ETC-642	New Zealand White rabbits HCAECs	↘ ICAM-1 and VCAM-1 in the artery wall ↘ monocyte adhesion ↘ NF-κB activation and VCAM-1 expression
[184]	PC/PS-rHDL	Macrophages LDLR^−/−^ mice	↘ IL-6 secretion ↘ circulating IL-6
[185]	4F	Primary monocytes THP1	↘ CCL2, IL-6 and TNF-α ↘ monocyte adhesion
[58]	D-4F	HCAECs	↘ HDL oxidation induced by H_2_O_2_
[186]	CER-001	HUVECs LDLR^−/−^ mice	↘ CCL2, IL-6 and IL-8 ↘ VCAM-1 in plaques
[187]	CSL-112	Small remodeled HDLs + PBMCs	↘ IL-6, IL-1β and TNF-α
[188]		Human whole blood	↘ ICAM-1 on monocytes and neutrophils ↘ IL-6, IL-1β and TNF-α
[189]	5A/PLPC	New Zealand White rabbits HCAECs	↘ ICAM-1 and VCAM-1 in the artery wall ↘ ROS production and NADPH oxidase expression ↘ NF-κB activation, ICAM-1 and VCAM-1 expression ↘ ROS production
[190]		C57BL/6 mice	↘ plasma IL-6, TNF-α, and CCL2 ↘ CD11b on monocytes ↘ in vivo leukocyte recruitment

“↘” and “↗” means “decrease” and “increase”, respectively. HCAECs, human coronary artery endothelial cells; HMDMs, human monocyte-derived macrophages; LDLR, low-density lipoprotein receptor; PC, phosphatidylcholine; PL, phospholipid; PS, phosphatidylserine.

**Table 2 antioxidants-13-00057-t002:** Summary of antioxidant and anti-inflammatory functions of HDLs and compositional alterations of HDLs in type 1 and type 2 diabetes.

Functions and Composition	Type 1 Diabetes	Ref.	Type 2 Diabetes	Ref.
Antioxidative properties	↘ LDL oxidation	[47]	↘ LDL oxidation ↘ lipid hydroperoxide removal ↘ ROS production	[52,55,56] [42] [57,58]
Anti-inflammatory properties	=adhesion molecule inhibition	[93]	↘ or =adhesion molecule inhibition ↘ monocyte migration ↘ TNF-α and IL-1β	[62,72] [103] [52] [100,101]
NO production	↘ NO production	[49]	↘ NO production	[57]
HDL proteome	PON-1 activity?	[47,49]	↘ PON-1 activity	[59,60,61,62]
HDL lipidome	↗ triglycerides ↘ S1P in HDL3 ↘ plasmalogens in HDL2	[50] [50]	↗ triglycerides ↘ plasmalogens	[105] [72]
HDL size	↗ large HDLs	[195]	Shift towards small HDL3	[105]

“↘”, “↗”, and “=” mean “decreased”, “increased”, and “similar” effects, respectively.

## Data Availability

The data presented in this study are available within the article.

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
