# Peer review of "Antioxidant and Anti-Inflammatory Functions of High-Density Lipoprotein in Type 1 and Type 2 Diabetes"

_antioxidants, 2023, doi:10.3390/antiox13010057_

Round 1

Reviewer 1 Report

Comments and Suggestions for Authors

This is a very nicely written and very comprehensive review on the antioxidant and anti-inflammatory functions of HDL in type 1 and type 2 diabetes. The review is another valuable contribution of the author to the field of HDL biochemistry/physiology.

There are some minors that need to be addressed:

Lane 191: …an acute-phase protein whom…  please replace ‘’whom’’ with ‘’whose’’

Lane 267: …and S1PR to…  ‘’via’’ might be more suitable than ‘’to’’

Lane 274: …, resulting to a.. please replace ‘’to’’ with ‘’in’’  or replace ‘’resulting’’ with ‘’leading’’

Lane 330: …are predictive OF all-cause mortality as well as with   cardiovascular-specific… (please add the missing OF and delete ‘’with’’.

Lanes 519, 520: …, but such they have…  not clear, please re-writte.

Lane 554: ….effective than… please replace ‘’than’’ with ‘’as’’.

Comments on the Quality of English Language

Fine, minor corrections needed.

Author Response

This is a very nicely written and very comprehensive review on the antioxidant and anti-inflammatory functions of HDL in type 1 and type 2 diabetes. The review is another valuable contribution of the author to the field of HDL biochemistry/physiology.

There are some minors that need to be addressed:

I thank the Reviewer for his positive feedback and comments. All corrections in the revised manuscript are in red color.

Lane 191: …an acute-phase protein whom…  please replace ‘’whom’’ with ‘’whose’’

I replaced the word “whom” with “whose” (line 238 in the revised manuscript).

Lane 267: …and S1PR to…  ‘’via’’ might be more suitable than ‘’to’’

I made the change suggested by the Reviewer (line 315 in the revised manuscript).

Lane 274: …, resulting to a.. please replace ‘’to’’ with ‘’in’’ or replace ‘’resulting’’ with ‘’leading’’

I replaced “resulting” with “leading” (line 320 in the revised manuscript).

Lane 330: …are predictive OF all-cause mortality as well as with cardiovascular-specific… (please add the missing OF and delete ‘’with’’.

I added “of” and deleted “with” (line 386 in the revised manuscript).

Lanes 519, 520: …, but such they have…  not clear, please re-writte.

I rephrased as follows: “[…] omega-3 fatty acids reduce triglyceride levels in patients with T2D [171], but such they also exert have pleiotropic effects on multiple atherosclerotic processes including oxidative stress and inflammation” (line 583 in the revised manuscript).

Lane 554: ….effective than… please replace ‘’than’’ with ‘’as’’.

I rephrased the sentence as follows: “[…] were as effective as HDL3 in inactivating lipid peroxides in LDLs” (line 617 in the revised manuscript).

Reviewer 2 Report

Comments and Suggestions for Authors

The review seems to be based on the common sense. On the other hand, several concerns could be raised for the improvement of the review.

1.      The recent reviews (for instance, Chapman MJ. HDL functionality in type 1 and type 2 diabetes: new insights. Curr Opin Endocrinol Diabetes Obes. 2022; Ganjali S, et al. HDL functionality in type 1 diabetes. Atherosclerosis. 2017) are cited, by which the difference in the viewpoints between the reviews could be discussed.

2.      Comparative discussion of the difference of HDL functionality between type 1 and type 2 diabetes, showing some different pathology and clinical course, would be expected.

3.      HDL particles carry many protein molecules; thus, the roles of not only PON1 but other molecules should be included in the review.

4.      The roles of not only Apo A-I and E but other apoproteins such as ApoA-II should be included in relation to the HDL size in the review.

5.      Regarding the HDL size, which size can be more antioxidative and anti-inflammatory? This should be more detailed.

6.      The knowledge of HDL efflux capacity could be more included.

7.      The mechanistical pathology and characteristic of glycoxidation should be detailed in introduction with any Figure.

8.      This review was based on the narrative way. The standard of selection of the cited literature could be indicated.

9.      The knowledge cited from Ref 134 was studied in metabolic syndrome, not diabetes.

10.   The knowledge cited from Ref 137 was studied in obesity, not diabetes.

11.   Row 56; the sentence might have some references.

12.   Row 62; the sentence might have some references.

13.   Row 118; the expression ‘paraoxonase 1’ or ‘paraoxonase-1’ (for instance, in row 98) was mixed in the text.

14.   Row 376; the expression ‘apoAI’ or ‘apoA-I’ was mixed in the text.

15.   Conclusive remarks should be stated in the end of text.

Comments on the Quality of English Language

Native check by trained speakers is again required in submission.

Author Response

The review seems to be based on the common sense. On the other hand, several concerns could be raised for the improvement of the review.

I thank the Reviewer for his comments. All corrections in the revised manuscript are in red color.

1) The recent reviews (for instance, Chapman MJ. HDL functionality in type 1 and type 2 diabetes: new insights. Curr Opin Endocrinol Diabetes Obes. 2022; Ganjali S, et al. HDL functionality in type 1 diabetes. Atherosclerosis. 2017) are cited, by which the difference in the viewpoints between the reviews could be discussed.

The two reviews by Chapman et al. and Ganjali et al. were not cited in the initial version of the manuscript. Firstly, the references of these two reviews were added in the revised manuscript (line 60) as suggested in response to the Reviewer’s comment #11 below: “On one hand, diabetes has detrimental effects on the composition and functions of HDLs, impairing, at least in part, their antiatherogenic properties [7–9].”. Secondly, I discussed the findings between the reviews in the discussion section (lines 625-628).

2) Comparative discussion of the difference of HDL functionality between type 1 and type 2 diabetes, showing some different pathology and clinical course, would be expected.

I added the Table 2 in the discussion section to compare the HDL functionality in type 1 and type 2 diabetes, in relation with HDL proteome and lipidome (lines 636-638). The gaps in the understanding of HDL functionality in each type of diabetes were further discussed in the discussion section (lines 666-672).

  1. HDL particles carry many protein molecules; thus, the roles of not only PON1 but other molecules should be included in the review.

The initial version of the manuscript addressed the roles of HDL proteins other than PON1, including apoA-I, apoM, CETP and glutathione peroxidase. As recommended by the Reviewer, the revised manuscript also delves into the role of other proteins, such as apoA-II and apoE. In particular, the potential role of apoA-II and apoE in the antioxidant functions of HDLs was added (lines 143-149). In addition, new insights into the potential contribution of apoE to the anti-inflammatory function of HDLs have been incorporated at lines 325-329.

  1. The roles of not only Apo A-I and E but other apoproteins such as ApoA-II should be included in relation to the HDL size in the review.

The role of apoA-I (lines 140-142 and 315), apoE (lines 146-149, 162-164, and 325-329) and apoA-II (lines 143-146, and 247-249) in the antioxidant and anti-inflammatory functions of HDLs are now mentioned and discussed regarding the HDL size (lines 179-186).

  1. Regarding the HDL size, which size can be more antioxidative and anti-inflammatory? This should be more detailed.

I mentioned in the revised manuscript the impact of HDL size on their antioxidant activity, in relation to distinct proteomic and lipidomic profiles (lines 179-186). I also did so for their anti-inflammatory properties (lines 323-329).

  1. The knowledge of HDL efflux capacity could be more included.

The consideration of HDL efflux capacity may seem beyond the scope of this special issue of Antioxidants, which is dedicated to exploring the antioxidant and anti-inflammatory functions of HDLs. Nevertheless, HDL-mediated efflux could potentially exert an indirect influence on the antioxidant and anti-inflammatory properties of HDLs, primarily by virtue of its involvement in endothelial NO production. The binding of HDLs to SR-BI via apoA-I causes cholesterol efflux that is sensed by SR-BI, leading to PDZK1-dependent activation of Src family kinases and Akt, which phosphorylates eNOS at Ser1177 and thereby increases NO production. Ultimately, as mentioned in the initial version of the manuscript, NO plays a role for the antioxidant and anti-inflammatory functions of HDLs. This discussion was included in the discussion section (lines 649-659).

  1. The mechanistical pathology and characteristic of glycoxidation should be detailed in introduction with any Figure.

The Figure 1 was added in the introduction section to summarize the mechanisms of lipoprotein glycoxidation (lines 73-87).

  1. This review was based on the narrative way. The standard of selection of the cited literature could be indicated.

A method section has been incorporated to describe the strategy employed for the literature search (lines 88-95).

  1. The knowledge cited from Ref 134 was studied in metabolic syndrome, not diabetes.

The mean fasting glucose was 8.18 ± 2.48 mmol/L in the study population of Sang et al. (ref. 134 in the initial version of the manuscript and 144 in the revised manuscript), suggesting that most of them had diabetes. Although not strictly conducted in a study population recruited based on the presence of diabetes, such study remains informative as exercise is an intervention recommended both in patients with metabolic syndrome and those with diabetes. The sentence was modified in the revised manuscript to better emphasize that the study was conducted in patients with metabolic syndrome and not strictly in patients with diabetes (lines 508-515): “Regarding the effects of lifestyle interventions on HDL anti-inflammatory function, HDL3 from patients with metabolic syndrome were more efficient in downregulating the expression of VCAM-1 on endothelial cells and the release of CCL2 after a 10-week walk/run training program in a study population not strictly recruited based on the presence of diabetes but rather on the presence of metabolic syndrome [144].”

  1. The knowledge cited from Ref 137 was studied in obesity, not diabetes.

The study by Roberts et al. (ref. 137 in the initial version of the manuscript and ref. 147 in the revised manuscript) enrolled overweight or obese men. Such lifestyle intervention study based on diet and exercise is informative since the large majority of patients with type 2 diabetes were overweight or obese, and both diet and exercise are common interventions in obesity and type 2 diabetes. The sentence was modified in the revised manuscript to better emphasize that the study was conducted in overweight or obese men and not strictly in patients with type 2 diabetes (lines 512-515): “A short-term diet and exercise intervention enhanced the capacity of HDLs to reduce monocyte chemotactic activity in obese men in a study that enrolled men based on the presence of overweight or obesity, rather than strictly on the presence of T2D [147].”

  1. Row 56; the sentence might have some references.

I cited three reviews focusing on the effects of diabetes on the composition and functions of HDLs (line 56 in the initial manuscript and line 60 in the revised manuscript): “On one hand, diabetes has detrimental effects on the composition and functions of HDLs, impairing, at least in part, their antiatherogenic properties [7–9].”.

  1. Row 62; the sentence might have some references.

I cited two reviews focusing on the antidiabetic effects of HDLs (line 62 in the initial manuscript and line 66 revised manuscript): “[…] HDLs seem to fulfill functions on glucose metabolism that mitigate the development and progression of diabetes [12,13].”

  1. Row 118; the expression ‘paraoxonase 1’ or ‘paraoxonase-1’ (for instance, in row 98) was mixed in the text.

I thank the Reviewer for having identified this error in the manuscript. Therefore, I standardized the word “paraoxonase-1” in the writing (line 155 in the revised manuscript).

  1. Row 376; the expression ‘apoAI’ or ‘apoA-I’ was mixed in the text.

The writing of the word “apoA-I” is now standardized (line 435 in the revised manuscript).

  1. Conclusive remarks should be stated in the end of text.

A conclusion section was added (lines 689-693), and the discussion section was reinforced.

Native check by trained speakers is again required in submission.

An English-native speaker proofread the revised manuscript. Several changes were made throughout the manuscript (in red color).

Reviewer 3 Report

Comments and Suggestions for Authors

This is an interesting and comprehensive review focused on the protective role of HDL in both diabetes and in the setting of atherosclerotic cardiovascular diseases. It discusses the contributions of these HDL functions to the onset of both type 1 diabetes and type 2 diabetes. The manuscript is well-written covering wide range of literature related to the field. Thus, I believe that the manuscript can be accepted for publication in Antioxidants.

I recommend to the authors to include a Figure or Table summarizing the alterations in the anti-oxidative properties of HDL in diabetes.

I also suggest discussing the anti-inflammatory role of HDL against the effect promoted on cells by modified LDLs. In this regard, the partly loss of protective action of HDL from type 2 diabetic patients against electronegative LDL should be mentioned.

Author Response

This is an interesting and comprehensive review focused on the protective role of HDL in both diabetes and in the setting of atherosclerotic cardiovascular diseases. It discusses the contributions of these HDL functions to the onset of both type 1 diabetes and type 2 diabetes. The manuscript is well-written covering wide range of literature related to the field. Thus, I believe that the manuscript can be accepted for publication in Antioxidants.

I thank the Reviewer for his comments. All corrections in the revised manuscript are in red color.

I recommend to the authors to include a Figure or Table summarizing the alterations in the anti-oxidative properties of HDL in diabetes.

The Table 2 was added to summarize the impairments in the antioxidative and anti-inflammatory properties of HDLs in type 1 and type 2 diabetes (lines 636-638).

I also suggest discussing the anti-inflammatory role of HDL against the effect promoted on cells by modified LDLs. In this regard, the partly loss of protective action of HDL from type 2 diabetic patients against electronegative LDL should be mentioned.

I agree with the Reviewer that the pro-inflammatory effects of modified LDLs were not discussed enough in the initial version of the manuscript, and subsequently the relevance of the anti-inflammatory role of HDLs regarding the deleterious effects of modified LDLs. In the revised manuscript, I further present the effects of oxLDLs, and therefore the anti-inflammatory role of HDLs against these particles (lines 114-120). The Figure 3 and its caption presented the impact of HDLs on the modified LDL-mediated effects on NF-κB pathway (lines 295 and 298-305). In addition, I mentioned in the revised manuscript the potential contribution of the loss of protective action of HDLs against LDL oxidation in patients with type 2 diabetes, regarding the anti-inflammatory properties of HDLs (lines 407-410).      

Reviewer 4 Report

Comments and Suggestions for Authors

The author reviewed the antioxidant and anti-inflammatory functions of high-density lipoproteins (HDLs) in type 1 and type 2 diabetes, and how they affect the risk of cardiovascular disease and the onset of diabetes. Specific comments:

1.          The abstract is well-written and concise, but it could be improved by adding a sentence about the implications or applications of the review. For example, how could the findings of this review inform future research or clinical practice?

2.          The discussion section is well-written and summarizes the main findings and limitations of the review. However, it could be improved by adding some suggestions for future research directions or clinical implications. For example, how could the measurement of HDL functions be used to improve the diagnosis, prognosis, or treatment of patients with diabetes and cardiovascular disease? What are the potential targets or strategies to modulate HDL functions in these populations? What are the knowledge gaps or challenges that need to be addressed in future studies?

Author Response

The author reviewed the antioxidant and anti-inflammatory functions of high-density lipoproteins (HDLs) in type 1 and type 2 diabetes, and how they affect the risk of cardiovascular disease and the onset of diabetes.

Specific comments:

I thank the Reviewer for his comments. All corrections in the revised manuscript are in red color.

  1. The abstract is well-written and concise, but it could be improved by adding a sentence about the implications or applications of the review. For example, how could the findings of this review inform future research or clinical practice?

I have appended a sentence to the end of the abstract to inform on the primary gaps that necessitate exploration in future research studies (lines 23-27).

  1. The discussion section is well-written and summarizes the main findings and limitations of the review. However, it could be improved by adding some suggestions for future research directions or clinical implications. For example, how could the measurement of HDL functions be used to improve the diagnosis, prognosis, or treatment of patients with diabetes and cardiovascular disease? What are the potential targets or strategies to modulate HDL functions in these populations? What are the knowledge gaps or challenges that need to be addressed in future studies?

I thank the Reviewer for these suggestions to improve the discussion section. I added some paragraphs in the discussion section to inform on gaps to address in future research studies regarding the pathophysiology of HDL functionality (lines 666-672), the assessment of cardiovascular risk in patients with diabetes (lines 673-682), and the impact of antidiabetic drugs on HDL functions (lines 686-688).

Round 2

Reviewer 2 Report

Comments and Suggestions for Authors

The paper was well revised. Row 31-3 (the first sentence of Intro) could have the reference cited.

Author Response

I thank the Reviewer for his positive feedback. I added the reference for the first sentence of the introduction section (lines 31-33).